# Design and Experimental Investigations of Shape and Attitude Carding System for the Wires of Micro Coreless Motor Winding

**DOI:** 10.3390/mi12101140

**Published:** 2021-09-23

**Authors:** Yuezong Wang, Liuqian Wang, Jiqiang Chen

**Affiliations:** Faculty of Materials and Manufacturing, Beijing University of Technology, Beijing 100124, China; yaozongw@bjut.edu.cn (Y.W.); chenjiqiang2021@163.com (J.C.)

**Keywords:** coreless motor, winding, robotics, micro-manipulation, brush-based micro-manipulator

## Abstract

The shape and attitude (S&A) of the electrode wire are important characteristics of micro coreless motor winding. The purpose of this paper is to present the design of a robotic micro-manipulation system for micro wire carding with arbitrary S&A, which can be used as the pretreatment system for wire micro-gripper systems. The system is based on the principle of flexible carding, and uses nylon, bristle, nanometer-silk and wool as materials for the brushing micro-manipulator. The trajectory of the brushing micro-manipulator is designed, and the S&A of the electrode wires are straightened through the combined motion mode of horizontal and vertical brushing micro-manipulators. The experimental results show that the material of the brushing micro-manipulator has a great impact on the carding quality. Nanometer-silk material is more suitable for horizontal brushing micro-manipulators, and wool material is more suitable for vertical brushing micro-manipulators. The geometric dimension of the brushing micro-manipulator also affects the carding quality. When the diameter is in the range of 1 mm, the carding effect of the horizontal brushing micro-manipulator with a length of 4.9–8 mm is better. The system can realize the automatic carding of flexible electrode wires with arbitrary S&A, and it will not damage the structure of wires in the process.

## 1. Introduction

The micro coreless motor has the advantages of compact size, small moment of inertia, high speed, fast response, smooth output torque and good servo characteristics, etc. [1], and it has been widely used in many fields, such as intelligent robotics, micro-efficient medical devices, smart cameras and intelligent mobile phones. In the manufacturing process of micro coreless motors, a variety of automation technologies related to manufacturing have appeared, such as winding technology [2], electrode wire automatic welding technology [3], motor integrated assembly technology [4], etc. The diameter and length of the electrode wire of the micro coreless motor winding are only tens of microns and several millimeters, and the shape and attitude (S&A)of the electrode wire produced are arbitrary. In the process of welding the electrode wire to the winding pad, the automatic wire micro-gripper robot, which automatically captures the electrode wire and pulls it to the position of the pad, is very important to realize the welding automation. For the wire micro-gripper robot [5,6], it is of great significance to improve its accuracy and stability through the use of pre-processing technology for the wire’s S&A, and make it meet the requirements of linearization and angle standardization.

Robot technology for wire micro-clamping and welding of flexible wires is often used in industry, such as gold wire bonding technology in semiconductor chips [7,8]. Gold wire is transferred to the heated end of the welding gun through a wire feeding mechanism for melting and welding to achieve the interconnection between pins of components or between pins of components and pins of peripheral chips. Welding techniques for wires are also common [9,10], where wires are transferred to gun and pad positions by wire feeding mechanisms for melting and welding. In this type of technology, the wire is passed forward by the wire feeding mechanism, the welding end of which is always held by the wire feeding mechanism and can be used as long as the wire is passed forward. In this paper, the welding problem of the electrode wire of the micro coreless motor winding that we studied is different from that of the welding technology used for wire feeding mechanisms. The electrode wire of the micro coreless motor winding is suspended, its length is about several millimeters, its S&A are not single after cutting and it shows randomness. Therefore, it is difficult to realize automatic welding by using the above technology. Before the welding, it is often necessary to pre-process its S&A, and then capture the suspended end using an automated wire micro-gripper robot to pull it to the welding pad position for welding. Manual operation is very common in the existing welding methods for the electrode wire of micro coreless motor windings. Operators use microscopy and a gripper to pull the wire manually and attach it to the welding pad. However, the manual pulling is inefficient and has poor consistency. In recent years, there has been some research on automatic wire-pulling methods for winding electrode wires of micro coreless motor winding. Lei and Liu et al. tried to blow the suspended end of the electrode wire to the pad position with the help of air-fluid [11,12], and change the posture of the electrode wire by adjusting the direction of the airflow. Wang et al. designed an automatic wire-pulling robot system based on the micro clamping principle, identified and monitored the position of the suspended end of the electrode wire through computer vision technology, clamped it with a micro-gripper, and then pulled the suspended end of electrode wire to the pad position for welding [13,14]. One precondition for these two methods is that the shape of the electrode wire should be as vertical as possible to ensure the accuracy of visual identification and suspension capture. However, the implementation of these two methods is very difficult for electrode wires with arbitrary S&A (such as curling, climbing, etc.). Therefore, it is very important for the wire micro-clamping technique to pre-process the initial S&A of the electrode wire so that its shape approximates that of a straight line and its attitude meets certain angle requirements.

After the winding of the micro coreless motor is made, the S&A of the electrode wire are random and arbitrary. For this type of winding, a flexible robotic micro-manipulation system is designed to pre-process the S&A of the electrode wire. In this system, the electrode wire is brushed by a flexible carding principle so that the S&A of the wire can meet the requirements of automatic wire-clamping. The system uses flexible material as the micro-manipulator, and a carding pattern of S&A is established by horizontal and vertical brushing. The flexible mechanism and the principle of flexible carding have been used in industry [15,16]. Shen et al. designed an agricultural harvester for oil-tea fruit, which uses flexible materials at the front and has a comb-brush structure. This flexible brush-based harvester can effectively avoid damage to the bud while harvesting fruit [17]. Aiming at the rotation problem of artificial satellites, Sun et al. proposed a rotary elimination mechanism based on a flexible deceleration brush. The momentum of the rolling target is dissipated by the contact collision between the end flexible deceleration brush and the rolling target windsurfing board [18]. In planetary science, sampling brushes are usually used in missions involving the extraction of small bodies, and this flexible contact method has been shown to be very effective [19]. Wang et al. proposed a straightening device for the micro flexible wire of micro-winding, which straightens the curved wire on the winding surface through the contact between the brush and the target wire [20]. However, this device does not consider the influence of different materials, structures and geometric dimensions on the flexible brush, resulting in low brush efficiency and unsatisfactory carding effect. The above research applies the principle of flexible carding, comb or brush structure designs, while completing the task to the greatest extent, avoiding destruction and interference with other objectives, and it reflects the flexibility and safety of flexible carding. Based on the work of Wang et al., a micro-manipulation system based on flexible carding is designed in this paper. The system is used to pre-process the S&A of the electrode wire of the micro coreless motor winding. It has the following advantages: (1) The flexible brush used as the micro-manipulator will not destroy the structure of the electrode wire and winding support, nor will it pull off the electrode wire. (2) The brushing micro-manipulator is composed of several brush units. The wire can be pulled by the gap between the brush units, and the arbitrary S&A of the wire can be carded by the gap. (3) It has very strong adaptability and is suitable for all shapes and attitudes, such as climbing, side-sticking and curling. After the S&A of electrode wire are pre-processed by the system, its position becomes relatively fixed, and then it enters the automatic wire-pulling system, which can ensure that the wire micro-gripper robot can grasp the wire in a relatively fixed space, thus improving the accuracy and stability of clamping to a certain extent.

The remainder of this article is organized as follows: in Section 2, the materials and methods are designed. In Section 3, the experiments and discussion are carried out. In Section 4, the conclusions are derived.

## 2. Materials and Methods

### 2.1. Analysis of Design Requirements

The winding is the core component of coreless motors. Its structure is shown in Figure 1a. It is composed of a winding, a winding bracket, an electrode pad, a shaft and electrode wires. The top of the winding usually contains 3~5 electrode wires, and the corresponding 3~5 pads are usually set at the top of the inner winding bracket. The top of the electrode wire needs to be welded onto the pad. The electrode wire of the coreless motor winding needs a series of processes before the winding can be finally assembled into the motor. In this paper, the technological process of automatic production is given as shown in Figure 1b–e. Figure 1a is an ideal winding structure with the electrode wires in a nearly vertical position; however, the electrode wires, in fact, have arbitrary S&A after winding of the coil, as shown in Figure 1b. The electrode wires of arbitrary S&A can approach the vertical state only after being straightened, as shown in Figure 1c. The electrode wires in Figure 1c are attached to the surface of the pad after being pulled by the micro-gripper robot system, as shown in Figure 1d. The wires in Figure 1d are then welded to the pad by an automated welding system, as shown in Figure 1e, and the winding is finally assembled into the micro coreless motor, as shown in Figure 1f.

The dimensions or dimension ranges of each part of the micro coreless motor winding in Figure 1a are shown in Table 1. The diameter of the electrode wire is in the order of tens of microns and belongs to a micro flexible body. At present, automatic processing has been realized in the technological process of welding (Figure 1d,e); however, in the wire-pulling session (Figure 1c,d), due to the small diameter, large flexibility and random S&A of the electrode wire, manual operation is common. The operator usually places one end of the wire on the welding pad manually by means of a microscope. Clearly, the efficiency of manual operation is low and its consistency is poor. In order to improve the automation of the wire-pulling process, it is important to establish a robotic system to enable automatic wire pulling.

We have designed an automatic wire-pulling robot system based on a micro-gripper [21], which realizes the automatic capture and pull operation of an electrode wire through the use of the micro-gripper. However, the S&A of the electrode wire must meet certain requirements before the wire-pulling system works stably, which is close to the S&A in Figure 1c, while the stable pulling of the electrode wire of arbitrary S&A in Figure 1b cannot be realized. Therefore, it is necessary to study the automatic carding technique of the S&A of the electrode wire before the automatic wire-pulling session.

The coil will be put into the container after winding. The S&A of the electrode wires are often changed and exhibit randomness, which generally manifests as bending and curling of the shape and has various attitudes. Therefore, the S&A can be concluded to conform to the basic shapes and attitudes in Figure 2 by dividing the attachment position and shape. Part or all of the electrode wires are suspended in the outer area of the winding in Figure 2a; focusing on the attitude, part or all of the electrode wires are attached or suspended on the surface of the welding pad in Figure 2b; furthermore, part or all of the electrode wires are attached to the outer sidewall of the winding in Figure 2c. The shape of the electrode wires in Figure 2a–c is straightened, while part or all of the wires in Figure 2d are bent and curled in shape. In practice, the attitudes and shapes in Figure 2a–d often mix to form more complex attitudes and shapes. From Figure 2, we can see that when the micro-gripper is used for pulling random S&A wires, it is likely to have problems related to clamping leakage, mistaken clamping and clamping stability, which will greatly reduce the stability of the whole system. To solve this problem, this paper studies the random S&A carding of electrode wire, and a robotic system is designed for the carding of small- diameter flexible wire, which can card the random attitude to a certain angle range and has a better shape, and helps to guarantee the success rate of the wire-pulling session through the pre-processing of wire S&A.

For the robotic system of S&A carding, the S&A of electrode wires, after carding, should meet certain conditions, as shown in Figure 2e, which can be summarized as follows:

Req1: The shape of the electrode wire is straightened. In Figure 2a–d, the initial state of the electrode wire may be folded, curled or greatly bent. After S&A carding, the shape of the electrode wire forms an approximate line with smaller curvature.

Req2: The attitude of the electrode wire should be in an angle range. When the condition in Req1 is satisfied, the main body of the electrode wire is also required to be at a conical angle ±*ω*. Ensure that the electrode wire is in an approximate vertical state within the range, and *ω* can be set according to actual conditions.

### 2.2. Design of Micro-Manipulation System

The structure of the designed electrode wire S&A carding system is shown in Figure 3a, including the micro-manipulator module, motion system, image acquisition system, load-supporting table, control system and computer. The micro-manipulator module is used in the arbitrary S&A carding of the electrode wires so that they meet the conditions in Req1 and Req2. The module consists of a horizontal brushing micro-manipulator, a vertical brushing micro-manipulator and a position–attitude adjustment mechanism with multiple degrees of freedom. The principle of flexible carding is used to deal with the S&A of the electrode wire. As shown in Figure 3b, two small-diameter flexible brushes are used in the micro-manipulator (see Section 2.2.1). One brush moves horizontally and the other brush moves vertically. The S&A of the electrode wire is carded by sequential brushing of two brushing micro-manipulators. The small-diameter electrode wire has a certain flexibility and can withstand small external force (in the mN order of magnitude). The principle of flexible carding cannot destroy the electrode wire, but also make the S&A of the electrode wire meet the requirements of Req1 and Req2. The motion of the micro- manipulator module is achieved by a motion system consisting of several small slider cylinders.

The image acquisition system consists of two sets of microscopic visual systems, each of which consists of a microscope and a camera. In Figure 3a, the microscopic visual system A is horizontally positioned. It is mainly used for monitoring the position of electrode wires, and for the vertical positioning of microscopic visual system B, as well as for image capturing and image analysis. The load-supporting table consists of an electronic control turntable and a special fixture. During the carding process, the winding is placed above the clamping mechanism, which can rotate in a horizontal plane. The control system is mainly used for motion control of the cylinder and the turntable, while the computer realizes the control of the whole system, as well as image acquisition and image processing.

The workflow of the system in Figure 3a can be broken down into the following steps:

Step 1: The winding is placed on the load-supporting table and the turntable starts to rotate at equal angles with a single rotation angle Ω, a minimum starting angle of 0° and a maximum rotation angle *θ*(*θ* ≥ 360°). For each rotation of the turntable, stop briefly at the current position and perform the following operations.

Step 2: After the turntable starts to move, the horizontal brushing micro-manipulator completes a cycle of movement on its own workstation according to the preset trajectory of motion for each Ω angle of rotation. When the turntable rotates to the *θ* angle, the horizontal brushing micro-manipulator completes the final movement and returns to the original position (see Section 2.2.2), at which point the turntable returns to its original position.

Step 3: After the turntable starts to move, the vertical brushing micro-manipulator completes a cycle of movement on its own workstation according to the preset trajectory of motion for each Ω angle of rotation. When the turntable rotates to the *θ* angle, the vertical brushing micro-manipulator completes the final movement and returns to the original position (see Section 2.2.2), at which point the turntable returns to its original position.

Step 4: When the turntable rotates to the maximum angle *θ*, the motion is stopped, the initial state is returned to, and the carding of the S&A of the electrode wire is completed.

#### 2.2.1. Micro-Manipulator Design

The brushing micro-manipulator used for S&A carding is an important part of the robot system. Based on the analysis of the mechanical characteristics of small-diameter flexible wire, the micro-manipulator module is designed based on the principle of flexible carding. In the process of the S&A carding of flexible wire, there is a complex interaction between the micro-manipulator and the flexible electrode wire. The interaction between the micro-manipulator and the flexible electrode wire can be regarded as an elastic contact model, which is used to analyze the elastic force between them. On this basis, the basic structure of the micro-manipulator is given, then the elastic forces of different materials are measured by means of mechanical experiments and used to optimize the geometric dimensions of the structure of the micro-manipulator. Based on these data, the geometrical parameters of the micro-manipulator that meet the requirements are given preliminarily. In general, the design of the micro-manipulator requires the determination of the material, structure and geometric parameters.

##### Flexible Carding Principle

In terms of the horizontal carding process, Figure 4a shows the state of the wire with the initial attitude just in contact with the horizontal brushing micro-manipulator. The flexible operating end of the brush bends due to the squeezing of the top of the winding, and the curled electrode wire at the top of the winding is randomly covered inside the brush units of the horizontal brushing micro-manipulator. Under the driving action, the brush generates a horizontal carding force *F*_1_ on the electrode wire and the friction force *f* between the wire and the winding surface. Then, when *F*_1_ > *f*, the brush produces a velocity on the electrode wire to change its original S&A. In this process, the horizontal carding force *F*_1_ plays a major role as the applied force, which is mainly due to the elastic contact between the brush and the electrode wire caused by the driving force. According to the Hertzian nonlinear elastic contact theory, they create an elastic contact force between them, which in turn exerts a force on the electrode wires. Figure 4b shows that the horizontal brushing micro-manipulator instantaneously leaves the top of the winding. Small displacement deformation occurs at the flexible end of the brush due to the release of the flexible end from the pressed state. Figure 4c shows the horizontal brushing micro-manipulator unfolding the wire horizontally after a small amount of displacement. The micro-manipulator applies the horizontal carding force *F*_3_ to the wire. The resulting horizontal direction guidance causes the shape of the wire to become straightened and the attitude to become horizontal.

In terms of the vertical carding process, Figure 5a shows the state of the wire with the initial attitude just in contact with the vertical brushing micro-manipulator. The brush contacts the winding sidewall and produces some minor deformation. Under the driving action, the micro-manipulator moves upwards, and its flexible operating end makes local elastic contact with the wire, applying carding force *F*_4_ vertically upward along the side wall of the winding. If the carding force is greater than the friction *f* between the winding surface and the wire, i.e., *F*_1_ > *f*, the electrode wire will change its original S&A under the guidance of the brushing micro-manipulator. Figure 5b shows that when the vertical brushing micro-manipulator is released from the side wall surface of the winding and reaches the top of the winding, a small displacement is generated by the recovery of the brush and an inclined carding force *F*_5_ is applied to the wire. Figure 5c shows the vertical upward carding force *F*_6_ generated by the micro-manipulator moving up the winding top after a small displacement. The carding force of the vertical brushing micro-manipulator is analyzed in three conditions with reference to horizontal carding.

##### Design of Brushing Micro-Manipulator

Small-diameter electrode wire has strong flexibility characteristics. It is curved in shape, specifically a complex curve. It not only bends, but also curls. Previous work has shown that the maximum tensile force that wires can withstand is in the order of tens of mN. Our research objective is to unfold the shape of the electrode wires through micro-manipulators and adjust their attitude to the specified angle range, which has special requirements for micro-manipulators. The principle of flexible carding is adopted in this paper. Using small-sized brushes as the operating end of the brushing micro-manipulator, and with the brush itself being a flexible body, soft materials such as nylon, bristle, nanometer silk (superfine fiber material) and wool are selected. The micro-manipulator does not destroy the electrode wires with reasonable geometrical dimensions, and the S&A of the electrode wires can be carded by means of the combined brushing effect of the brush units. The brushing micro-manipulator is designed in accordance with the following rules: first, the basic structure of the micro-manipulator is proposed; then, the materials required for the design of the micro-manipulator are selected; finally, the geometric dimensions of the micro-manipulator are determined.

The basic structure of the proposed brushing micro-manipulator is shown in Figure 6. It consists of a horizontal brushing micro-manipulator and a vertical brushing micro-manipulator. Their structure is shown in Figure 6a,b. In Figure 6a, the horizontal brushing micro-manipulator consists of a position–attitude adjustment mechanism with four degrees of freedom and a brusher. The position–attitude adjusting mechanism uses three degrees of translation and one degree of rotation, a combination of three dovetail guide rail translation modules and a yaw device to adjust the position and pitch angle of the micro-manipulator within a small range. The brusher consists of a base and flexible brush unit. Several brush units are glued together and placed into a thin tube of the base, with one end fixed and the other free floating. In Figure 6b, the vertical brushing micro-manipulator also consists of a position–attitude adjustment mechanism with four degrees of freedom and a brusher. Unlike the horizontal brusher, however, the vertical brusher is fixed at both ends to provide a more stable and focused contact force. Wires in different directions have different shapes and attitudes. A better carding effect can be achieved by selecting different brushing micro-manipulators. The main geometric dimensions of each component in Figure 6 are shown in Table 2.

We use organic materials to make brushes, which have the advantages of light weight, high specific strength, low density and good elasticity, among others. Nylon, bristle, nanometer silk and wool are selected as the materials of the brushes. The diameter of the brush units of these four materials are 0.15 mm, 0.21 mm, 0.05 mm and 0.03 mm, respectively. The geometric dimensions of the brushes mainly include diameters *Φ* and length *L*; the design of these two parameters takes into account both the size limitations of the target winding and the requirements of the mechanical properties. Clearly, the larger the overall diameter of the brush, the more brush units it contains. The interaction between the brush units and the electrode wire is very complex. Because the electrode conductor will be filled into the gap between the brush units, the S&A of electrode wire are changed along with the moving of the brush units. Therefore, the larger the diameter of the brush, the better the effect of carding. However, the diameter of the winding is only 1–1.5 mm. The brush with a larger diameter may interfere with other components and occupy a larger space. Brushes of smaller diameter take up less space and are not likely to collide with other components, so the smaller the brush diameter, the more secure the micro-manipulation is. In the process of designing the brush, the diameter of the brush should be increased as much as possible in an effective and safe workspace. Therefore, the value range of *Φ* is set to 0.5–1.0 mm to meet the above design rules to the greatest extent.

The vertical brush is fixed at both ends. Its length value has little influence on the quality of vertical carding. When determining the length value, we follow the principle of not colliding with other components during movement. Length *b* is set to 10 mm. The horizontal brush adopts the mode of fixing at one end and suspending at the other. Its length *L* will affect the carding performance of the brushing micro-manipulator. We determine the reasonable value of *L* through the elastic force test of different material brushes (see Section 3.1). The experimental results show that the theoretical design length *L* of the horizontal brush is 4.9–8 mm.

#### 2.2.2. Motion Control of Micro-Manipulator

The motion control principle of the micro-manipulator is shown in Figure 7. The winding, turntable, horizontal brushing micro-manipulator and vertical brushing micro-manipulator are arranged in an orderly manner according to their respective positions. The winding is placed on a special fixture, which is fixed on the electric control turntable, with the horizontal brushing micro-manipulator on the left and the vertical brushing micro-manipulator on the right. The two brushing micro-manipulators and the winding are adjusted to the same working plane (the *O*–*XZ* plane of Figure 7) by a position–attitude adjustment mechanism with four degrees of freedom. Both the horizontal and vertical brushing micro-manipulator are fixed on a two-axis motion system by a mechanical structure in the middle; each of the motion systems consists of two slider cylinders and can move point-to-point in both the X and Z directions. During the operation of the system, the electronic control turntable moves in the manner described in Step 1 above. For each rotation of the Ω angle, both the horizontal and vertical brushing micro-manipulator move in a cycle that is determined by the preset trajectory in order to achieve the horizontal and vertical carding.

To illustrate the trajectory and force of the brush, a sample coordinate system *O*–*XYZ* is established, in which the plane *O*–*XY* is located in the top plane of the winding, the origin *O* is located in the center of the circle where the central shaft intersects the plane, the *X*-axis points to the right, the *Z*-axis points upward, and the *Y*-axis is perpendicular to the *O*–*XZ* plane and points out of the paper.

The motion trajectory of the horizontal brush is shown in the left part of Figure 7. The brush is placed vertically. Based on the initial position, the trajectory of the horizontal brush consists of straight and curved segments, with a point T (*X*_T_, *Y*_T_, *Z*_T_) at the bottom of the brush as the reference point, and the trajectory of the horizontal brush is described by the movement of point T. Point T is above the plane *O*–*XY*, i.e., *Z*_T_ ≥ 0. When *Z*_T_ = 0, the brush just contacts the top plane, and the friction between the brush and the top plane is the smallest. When *Z*_T_ < 0, the friction between the brush and the top plane increases with the increase in |*Z*_T_|. The value of *Z*_T_ needs to be selected reasonably so that the electrode wire will not move downwards from the top plane. The motion of point T consists of the following trajectories:

Step 1: Moving along the trajectory section T–A, point T first moves to the preset point A, with point A being the starting point of the trajectory.

Step 2: Moving along the trajectory section A–B, point T moves from point A to point B and lifts forward along the *Z*-axis.

Step 3: Moving along the trajectory section B–C, point T moves horizontally from point B to point C, which is located near the winding central shaft.

Step 4: Moving along the trajectory section C–E, point E (*X*_E_, *Y*_E_, *Z*_E_) is located below the plane *O*–*XY*, where segment CE intersects with plane *O*–*XY* and point D. During the descent of the brush along the *Z*-axis, it is blocked by the top plane of the winding at point D and can no longer move along *Z*-axis to point E. Therefore, the brush deforms at point D and presses on the top plane of the winding. There is a pressure between the brush and the top plane of the winding. The pressure depends largely on the value of |*Z*_T_|.

Step 5: Moving along the trajectory curve D–F–G–H, point T moves horizontally from point D to point F, which is outside the top area of the winding. If there is an electrode wire in the section D–F area, the brush pulls the electrode wire to the outside of the winding using elastic contact force. When point T disengages from the force-applied plane, it follows the curve F–G for a small distance of curve motion, then moves horizontally in a straight line along segment G–H and returns to the starting point. In section F–H, the brush exerts a continuous force on the electrode wire, which will straighten the shape of the electrode wire.

Step 6: Moving along the trajectory section T–A, point T returns the initial position of point T in the horizontal direction from point H to complete the horizontal carding.

The motion trajectory of the vertical brush is shown in the right part of Figure 7. The brush is placed horizontally. Based on the initial position, the trajectory of the vertical brush consists of straight and curved segments, with a point I (*X*_I_, *Y*_I_, *Z*_I_) at the left of the brush as the reference point, and the trajectory of the vertical brush is described by the movement of point I. Point I is located below the top plane of the winding, i.e., *Z*_I_ < 0. The motion of point I consists of the following trajectories:

Step 1: Moving along the trajectory section I–J, point I first moves to the preset point J, with point J being the starting point of the trajectory.

Step 2: Moving along the trajectory section J–L, point L (*X*_L_, *Y*_L_, *Z*_L_) is located on the left side of the boundary line KM on the right side of the winding, where segment JL intersects with the boundary line at point K (*X*_K_, *Y*_K_, *Z*_K_). During the horizontal movement of the brush along the J–L section, it is blocked at point K by the right side of the winding and can no longer move negatively to point L along the *X*-axis. Therefore, the brush deforms at point K and compresses the right surface of the winding. There is pressure and friction between the brush and the right surface of the winding. The pressure and friction depend largely on the value of |*X*_K_–*X*_L_|. When |*X*_K_–*X*_L_| = 0, the brush just contacts the right side of the winding and gives zero pressure and friction to the winding surface. When |*X*_K_–*X*_L_| > 0, the friction between the winding and the brush increases with |*X*_K_–*X*_L_|.

Step 3: Moving along the trajectory curve segment K–M–N–O, point I moves from point K to point M along the *Z* direction. Point M is the intersection of boundary line KM on the right side of the winding and plane *O*–*XY*. If there is an electrode wire in the K–M interval, the brush pulls it above the *O*–*XY* plane using elastic contact force. When point I disengages from the force-applied plane, it follows the curve segment M–N for a small distance of curve motion and then moves straight up along the line segment N–O in the *Z*-axis direction, and the brush exerts a continuous force on the electrode wire, which will change its shape.

Step 4: Moving along the trajectory section O–P, point I moves horizontally from point O to point P.

Step 5: Moving along the trajectory section P–J, point I moves from point P along the negative direction of the *Z*-axis to point J.

Step 6: Moving along the trajectory section J–I, point I returns horizontally from point J to the initial position of point I to complete the vertical carding.

## 3. Results

In this section, the length of the brush of the horizontal micro-manipulator, designed through the use of brush elastic force measurement, and the S&A carding experiment, carried out using the system shown in Figure 3, are detailed. The S&A of the electrode wire were analyzed experimentally and the carding effects of brushing micro-manipulators of different sizes were compared. The X–Z two-axis motion system mainly used the vertical combination of MXQ8-20BS and MXQ6-10AT model slide cylinders. The air pump pressure was 0.6 MPa and the running speed was 0.4 m/s. The resolution of the camera used in the microscopic vision system was 1280 × 1024 pixels, with a 0.45×–4.5× magnification range, and it used microscopic vision systems A and B to take pictures of the carding results of the electrode wires from the side and top, respectively. The angular resolution of the turntable was 0.01° and the angle of a single rotation during the experiment was Ω = 1.0°.

### 3.1. Measurement of Brush Elastic Force

During the movement of the brush, it made close contact with the winding surface and compressed the surface. The contact part produces elastic force due to deformation. The greater the elastic force, the stronger the ability to pull the wire. However, flexibility needed to be considered in this process. This section estimates, by means of an experiment, the contact elastic force of the horizontal brush under bending deformation. Figure 8a gives the experimental principle. We fixed the brush unit on a base. The brush unit had both ends A and B, where point A was fixed and point B was suspended. We set the maximum length of the brush unit without deformation as *H*. We pulled the brush unit at the length *h* (*h* < *H*) using a dynamometer. Its contact point was N, the direction of tension *F* was in the horizontal direction, and the horizontal movement distance of contact N was Δ*x*. We measured the relationship between Δ*x* and the pulling force *F*. Figure 8b shows the established experimental system; the dynamometer was a high-precision digital push-pull meter with a range of 0–5 N, an accuracy of 1% of the measured value, and a sampling rate of up to 1000 times/s. The dynamometer was fixed on a set of the two-dimensional translation stage (with a grating ruler, the repeat positioning accuracy was 1 μm), which could move along the *X*-axis or *Y*-axis direction to adjust the position of the dynamometer. Figure 8c shows a partial enlargement of the brush pulled by the dynamometer. The brush unit was fixed on the base, the initial state remained vertical, and contact with the dynamometer was achieved through a special fixture. In the process of force measurement, we controlled the two-dimensional translation stage to ensure even movement to the right; the moving distance Δ*x* was provided by a grating ruler of the electrically controlled translation stage. The elastic contact force of the brush was approximately replaced by the pulling force *F*, which was measured and recorded by the dynamometer.

In the experiment, brushes were produced from brush units of nylon, bristle, nanometer silk and wool four materials. We produced four samples of each material and the length of all samples was *H* = 20 mm. In the experiment, the minimum value of *h* corresponding to the contact point N was 2 mm. When the translation stage driving dynamometer moved to the right, the value of Δ*x* gradually increased and we set the maximum Δ*x* value to 8 mm; then, we measured the force *F* at different Δ*x* values and plotted the *F*-Δ*x* scatter plot, as shown in Figure 9.

Analyzing the experimental data in Figure 9a–c, the elastic force *F* of nylon, bristle and nanometer silk always increased first and then decreased with increasing displacement Δ*x*. There was an obvious peak value in the *F*-Δ*x* relationship, which was the maximum elastic force of the brush, but the peak of *F* was slightly different between different samples of the same material and its corresponding Δ*x* was also different. As shown in Figure 9d, in the elastic force experiment of wool material, the *F*-Δ*x* scatter plot showed no smoothness other than that of the first three materials, and the overall elastic force was significantly small.

Table 3 shows the mean of Δ*x* and the mean of *F*-peak of all samples of different materials. Comparing the mean value of *F*-peak of different materials, the order of the mean *F*-peak values from big to small was bristle > nanometer silk > nylon > wool. When the *F*-peak appeared, the mean of Δ*x* corresponded to the range of 3.1–4.2 mm. Within this range, we determined the length range of the corresponding *h* (contact point N to fixing point A). By manual measurement, the length range was 4.9–8.0 mm, which is the theoretical length range with the greatest elastic force of the brush unit. Therefore, referring to the experimental results, the length of the horizontal brushing micro-manipulator was designed, taking 4.9–8.0 mm as our initial theoretical design length range and using it in the horizontal carding experiment (see Section 3.2) to verify its effect.

### 3.2. Horizontal Carding Experiment

The horizontal brushes were made of brush units of nylon, bristle, nanometer silk and wool. The diameter of all brushes was *Φ* = 1.0 mm and four brushes were made of each material. A total of sixteen models of brushes were used in the horizontal carding experiment to verify their carding effect. These sixteen models of horizontal brushes are shown in Table 4.

Sixteen windings were selected as test samples, each containing three electrode wires with random S&A, but as close as possible to the top surface of the winding. A brushing micro-manipulator carded a sample with the carding trajectory at the preset level, as shown in the left part of Figure 7. In order to assess the effect of horizontal carding on each wire, the S&A of the electrode wire after brushing should meet the following criteria, which are assessed according to subjective evaluation:

Req1_H: The shape of the electrode wire needs to meet the Req1 subjective criteria in Section 2.1.

Req2_H: The attitude of the electrode wire satisfies the attitude shown in Figure 2a, as far as possible, so that the later vertical carding can be carried out more effectively.

Req3_H: The S&A of each electrode wire has a good consistency after carding.

The Req1_H, Req2_H, Req3_H criteria can be divided into five evaluation levels, as shown in Table 5, of which 1–3 constitute the acceptable levels in terms of the carding effect, and 4 and 5 are the poor carding effect levels.

Table 6 shows the results of carding by using the brushing micro-manipulators made of nylon and bristle in the horizontal direction. In the experiment, each bush carded three electrode wires, identified as Wire A, Wire B and Wire C. Of the four nylon brushes (Brush(a–d)), for the carding result obtained using a 3 mm long brush (Brush(a)), only Wire C had a shape that met the Req1_H criterion, Wire B had an attitude the met the Req2_H criterion, and the consistency of the three wires did not meet the requirements of Req3_H. In the carding result obtained using a 5 mm long brush (Brush(b)) on the winding, all of the wires met the Req1_H criterion, and Wire A and Wire B met the Req2_H criterion, but the three wires showed interference in their carding and the consistency between the three wires did not conform to the Req3_H requirements. In the carding result obtained using a 7 mm long brush (Brush(c)) on the winding, all three wires did not only meet the Req1_H criterion in terms of shape but also met the Req2_H criterion in terms of attitude. However, the consistency between the three wires did not comply with Req3_H. In the carding result obtained using a 10 mm long brush (Brush(d)) on the winding, only the shape of Wire B met the Req1_H criterion, Wire A and Wire B had an attitude that met the Req2_H criterion, and the three wires were in the state of insufficient carding. The consistency of the three wires did not meet the Req3_H requirements. Of the four bristle brushes (Brush(e–h)), in the carding result obtained using a 3 mm long brush (Brush(e)), only the shape of Wire A met the Req1_H criterion but the attitude did not conform to Req2_H; the rest of the wires did not conform to Req1_H and Req2_H, and wire breakage occurred in the carding result of Wire C, which was not allowed. The consistency of the three wires did not meet the Req3_H requirements. In the carding result obtained using a 5 mm long brush (Brush(f)) on the winding, only the S&A of Wire A and Wire B met Req1_H and Req2_H; unfortunately, a small amount of tearing was present in the carding result of Wire C; this should be avoided. The consistency of the three wires did not conform to the Req3_H requirements. In the carding result obtained using a 7 mm long brush (Brush(g)) on the winding, all of the wires had a shape that met the Req1_H criterion and the attitude of Wire A and Wire B met the Req2_H criterion, but the consistency between the three wires did not comply with Req3_H. In the carding result obtained using a 10 mm long brush (Brush(h)) on the winding, only the S&A of Wire A and Wire B met Req1_H and Req2_H, and the consistency of the three wires did not meet the Req3_H requirements. Overall, the 7 mm length brush of the horizontal micro-manipulator was ideal for horizontal carding in nylon materials. Among the bristle materials, the 3 mm and 5 mm length brush of micro- manipulator broke wires during horizontal carding, and the 7 mm brush of micro- manipulator showed satisfactory carding results.

Table 7 shows the results of carding obtained by using the brushing micro-manipulators made of nanometer silk and wool in the horizontal direction. In the experiment, each bush carded three electrode wires, identified as Wire A, Wire B and Wire C. Of the four nanometer silk brushes (Brush(i–l)), in the carding result obtained using a 3 mm long brush (Brush(i)), the S&A of all three wires met the Req1_H criterion and the Req2_H criterion, but Wire B was in the state of carding interference. The consistency of the three wires did not meet the requirements of Req3_H. In the carding result obtained using a 5 mm long brush (Brush(j)) on the winding, the S&A of all three wires met Req1_H and Req2_H, and the consistency between the three wires complied with the Req3_H criterion. In the carding result obtained using a 7 mm long brush (Brush(k)) on the winding, the S&A of all three wires conformed to the Req1_H criterion and the Req2_H criterion, and the consistency between the three wires met the Req3_H criterion. In the carding result obtained using a 10 mm long brush (Brush(l)) on the winding, the shape of all three wires met the Req1_H criteria and the attitudes of Wire A and Wire B conformed to the Req2_H criterion, but the consistency between the three wires did not comply with the Req3_H criterion. Of the four wool brushes (Brush(m–p)), in the carding result obtained using a 3 mm long brush (Brush(m)), both Wire A and Wire C had an S&A that met the Req1_H and Req2_H criterion, but Wire B did not. The consistency between the three wires did not conform to the Req3_H requirements. In the carding result obtained using a 5 mm long brush (Brush(n)) on the winding, only Wire A had a shape that met the Req1_H criterion; Wire A and Wire C had an attitude that met the Req2_H criterion. The consistency between the three wires did not meet the Req3_H requirements. In the carding result obtained using a 7 mm long brush (Brush(o)) on the winding, the S&A of all wires did not conform to the Req1_H and Req2_H criterion, and the consistency between wires was more inconsistent with the Req3_H criterion. In the carding result obtained using a 10 mm long brush (Brush(p)) on the winding, the S&A of all wires did not meet the Req1_H and Req2_H criterion, and the consistency between wires also did not conform to Req3_H. As a whole, as far as nanometer silk was concerned, all the brushes of micro-manipulators with different lengths could achieve the target S&A carding more effectively, among which the carding results obtained using the 5 mm and 7 mm long brushes of micro-manipulators were the best. However, in wool materials, the results of horizontal carding with different lengths of brushes were not satisfactory.

As shown in Table 8, the horizontal carding experiment results were subjectively evaluated and compared according to the evaluation level in Table 5. The brush made of nanometer silk had an excellent overall carding effect, while the wire that was carded by wool material had the worst effect, and the bristle material brush broke the wire when it was of a shorter length; this situation should be avoided. The brushes of the horizontal micro-manipulators of different materials generally demonstrated the rule of carding interference at short length and insufficient carding at long length. In conclusion, the horizontal brushing micro-manipulator made of nanometer silk had the best carding effect in this experiment, and its carding was best when its length was 5 mm or 7 mm. The length of 5 mm and 7 mm also corresponded to our theoretical design length range of 4.9–8.0 mm in the experiment on the measurement of brush elastic force.

### 3.3. Vertical Carding Experiment

When making vertical brushes using the brush units of the four materials listed in Table 3 and one brush made of each material, all brushes were 10 mm in length *L* and 1 mm in diameter *Φ*. The four vertical brushes are shown in Figure 10. The carding effect was compared by using the vertical micro-manipulator with the four vertical brushes to card along the vertical direction.

Eight windings were selected as experimental samples and three electrode wires per winding sample. All samples were divided into two groups, each containing four windings. In the first winding group, all of the windings were carded by the horizontal brushing micro-manipulator and the shapes of the electrode wires were straightened horizontally, similar to the S&A of the wires in Figure 2a. The second winding group had the electrode wires that were as close to the outer wall of the winding as possible, or attached to it, similar to the S&A of the wires in Figure 2c. Vertical carding was performed with a vertical brushing micro-manipulator and the images of the carding results were taken by a microscopic visual system for analysis. We used each vertical brushing micro-manipulator, made of the vertical brush of different materials, to card a sample, and the brushing trajectory used was the preset trajectory shown in the right part of Figure 7. We evaluated the results of vertical carding with reference to the evaluation criteria shown in Table 5, where Req1_V, Req3_V and Req1_H, Req3_H were the same, and the content of Req2_V was changed according to requirements. The posture of the electrode wire needed to meet the Req2 subjective criteria outlined in Section 2.1.

Table 9 shows the results of carding the first and second winding groups using four vertical brushes (Figure 10a–d). In the experiment, each bush carded three electrode wires, identified as Wire A, Wire B and Wire C. For the first winding group with horizontal floating wires, in the carding result obtained using nylon brush (the brush shown in Figure 10a), Wire A and Wire B did not conform to the Req1_V criterion in terms of shape, while only Wire C met the Req1_V criterion, and all three wires, in terms of attitude, did not meet Req2_V. The three wires did not conform to Req3_V in terms of consistency. In the carding result obtained using bristled brush (the brush shown in Figure 10b), only Wire B had a shape that met the Req1_V criterion, and Wire A and Wire B met the Req2_V criterion in terms of attitude, but the consistency of the three wires did not meet the requirements of Req3_V. In the carding result obtained using nanometer-silk brush (the brush shown in Figure 10c), the S&A of all three wires met Req1_V and Req2_V, and the consistency between the three wires complied with the Req3_V criterion. In the carding result obtained using wool brush (the brush shown in Figure 10d), the S&A of all three wires also met Req1_V and Req2_V; similarly, good consistency existed between the wires, which conformed to the Req3_V criteria. For the second winding group with wires attached to the side wall, in the carding result obtained using brush(a), only Wire B had a shape that met the Req1_V criterion, and Wire A and Wire C met the Req2_V criterion in terms of attitude, but the consistency of the three wires did not meet the requirement of Req3_V. In the carding result obtained using brush(b), only the shape of Wire A met the Req1_V criterion and the attitude of Wire A and Wire B met the Req2_V criterion. The consistency of the three wires did not meet the Req3_V requirements. In the carding result obtained using brush(c), all three wires met Req1_V in terms of the shapes of the wires, but only the attitude of Wire B meets the Req2_V criterion. The consistency of the three wires did not meet the Req3_V requirements. In the carding result obtained using brush(d), the S&A of all three wires met Req1_V and Req2_V; however, the consistency between the wires was slightly poor and did not conform to the Req3_V criterion. As a whole, for the first group of winding vertical carding results, the vertical micro-manipulator with the brush made of nanometer silk and wool materials had an ideal carding effect and ensured that the S&A of the wires on the winding were consistent to a certain extent, while the micro-manipulator of the brush made of nylon and bristle materials had a poor effect on vertical carding. The carding results of the second group of windings showed that the vertical micro-manipulator with the brush made of wool material had the best carding effect, followed by nanometer silk material and nylon material. The brush made of bristle material had the worst vertical carding effect.

As shown in Table 10, the vertical carding results of the first and second winding groups were subjectively evaluated and compared according to the evaluation level shown in Table 5. It was found that, on the whole, the vertical brushing micro-manipulator designed in this paper could handle the horizontal floating state wire better than the electrode wire that was attached to the side wall. Vertical carding was more effective when the brush of the vertical micro-manipulator was made of wool material, and it also had a certain suitability for the windings in both cases.

### 3.4. Combined Carding Experiment

The effects of horizontal carding and vertical carding were verified by experiments in Section 3.2 and Section 3.3. It was found that the horizontal carding effect of the horizontal micro-manipulator with a brush that was made of 5 mm or 7 mm nanometer silk material was excellent, while the vertical carding effect of the vertical micro-manipulator with a brush that was made of wool material was better. In this section, we carried out the combined carding experiment, first with horizontal combing, then with vertical combing, and finally collected the image of carding results and analyzed the carding effect. A combined brushing micro-manipulator, used as an integrated machine, which consisted of a combination of horizontal and vertical micro-manipulators, was used in the experiment. The horizontal micro-manipulator with a brush was made of nanometer silk material with diameter of 1 mm and length of 7 mm. The vertical micro-manipulator with a brush was made of wool with a diameter of 1 mm and a length of 10 mm. We selected six windings as the experiment sample, each containing three electrode wires with arbitrary S&A. Six windings were carded separately using a combined brushing micro-manipulator. The carding trajectory used the preset carding trajectory shown in Figure 7.

Table 11 shows the results of the combined carding. The results show that the combined brushing micro-manipulator used in the experiment had an excellent effect in terms of carding the S&A of the electrode wire. The wire shapes of six samples of windings were all uncurled. In terms of attitude, part of the wires of Winding_2 was close to the outside of the winding and there was insufficient carding in terms of vertical carding, while part of the wires of Winding_4 and Winding_5 were close to the inner side of the winding and showed carding interference in terms of vertical carding. On the whole, the results for combined carding met our needs.

## 4. Discussion

In this paper, a robotic micro-manipulation system was designed to solve the problem of the micro electrode wire carding with arbitrary S&A, which exists in automatic wire-clamping using a micro-gripper robot system after the production of the coreless motor winding. The system consists of a micro-manipulator module, a motion system, an image acquisition system, a load-supporting table, a control system and a computer. It can realize the automatic carding of micro wires with arbitrary S&A, the shapes of the electrode wires are uncurled, and the attitudes of electrode wires are at a convenient angle for the micro-gripper clamping. Therefore, it is very suitable as a pretreatment system for wire micro-gripper robot systems. Based on this system, the arbitrary S&A carding of flexible micro electrode wire was studied. Firstly, a brushing micro-manipulator with different structures, materials and geometrical dimensions was designed. Then, the elastic force of various flexible materials was measured experimentally, and the carding performances of different micro-manipulators, for specific directions of electrode wires, were analyzed experimentally. Finally, the selected combined brushing micro-manipulator was used for carding the wires with arbitrary S&A, and the following conclusions were drawn:

(1) In the process of carding the S&A of electrode wire, the wire can withstand a force in the order of tens of mN. An excessive rigid force will damage the structure of the wire and affect the use of subsequent windings. Therefore, the principle of flexible carding should be adopted in the design of micro-manipulators to protect the structure of the wire while meeting the requirements.

(2) By measuring the elastic force of four kinds of flexible materials, it was found that the elastic force relationship of the four materials selected in this paper is as follows: bristle > nanometer silk > nylon > wool. Selecting different materials to produce the brush of the micro-manipulator will achieve different carding effects. According to the requirements of the operating objects in this paper and the elastic force measure experiment, the brush design length range in the horizontal direction should be 4.9–8 mm.

(3) It can be seen from the separate experiments of horizontal and vertical carding that the brushing micro-manipulator designed in this paper can card the micro electrode wires to a predetermined S&A. In order to avoid collision interference during carding, the design diameter of the brush of the micro-manipulator in both directions should be less than 1 mm. It was found through experiments that it has a better carding effect when the brush of the horizontal micro-manipulator uses nanometer silk material and the length is 5 mm or 7 mm. The brush of the vertical micro-manipulator works better with wool material and the length has no effect on vertical carding.

(4) The experiments show that the problem of the micro wire carding with arbitrary S&A can be solved by carding with the combination of horizontal and vertical micro-manipulators. This method does not need to pre-classify and re-process wires’ S&A manually, which greatly simplifies this tedious procedure. Therefore, automatic carding for the S&A of flexible micro wires is better realized through the combined carding operation, and the system designed in this paper is very suitable as a pretreatment system for wire micro-gripper robot system

It was found through experiments that the system designed in this paper does not achieve absolute straight linearity when carding wires with arbitrary shapes. Moreover, the curvature after carding is uncontrollable, and there is still significant room for improvement. During the carding process, it sometimes needs several repeated operations to achieve a better carding effect, and this periodic repetition can be avoided by optimizing the structure of the micro-manipulator. In addition, the position of the carded electrode wire cannot be accurately described, so the image processing can be used to describe, compare and analyze the accuracy.

## Figures and Tables

**Figure 1 micromachines-12-01140-f001:**
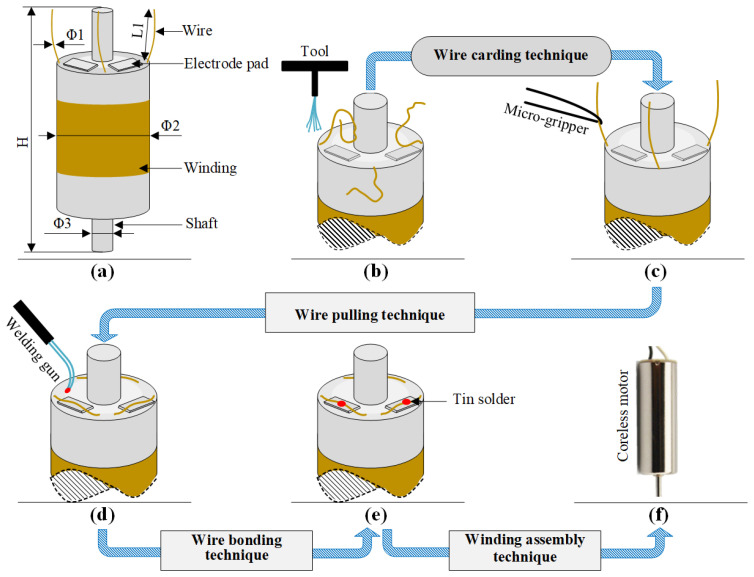
Coreless motor’s winding and its assembly process: (**a**) The winding of the coreless motor. (**b**) Electrode wires with random shape and attitude (S&A). (**c**) Electrode wires are carded to the target S&A through an automated wire carding technique. (**d**) Electrode wires are dragged and placed on the pads through an automated wire pulling technique. (**e**) Electrode wires are soldered through an automated wire bonding technique. (**f**) The final coreless motor is assembled through an automated winding assembly technique.

**Figure 2 micromachines-12-01140-f002:**
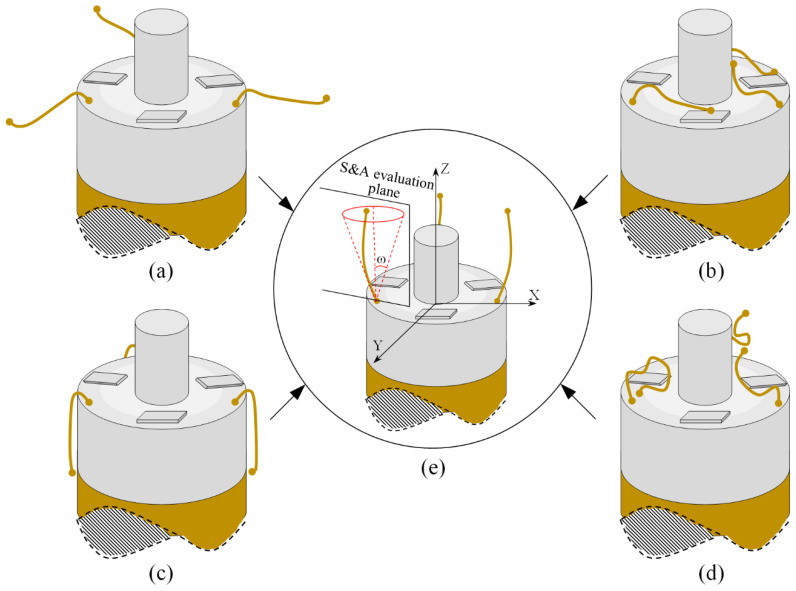
The S&A of electrode wires: (**a**) The attitude of the wires suspended on the outside of the coil. (**b**) The attitude of the wires attached or suspended on the pad surface. (**c**) The attitude of the wires attached to the periphery of the coil sidewall. (**d**) The shape of the wires with random curl. (**e**) The ideal brushing result of wire S&A.

**Figure 3 micromachines-12-01140-f003:**
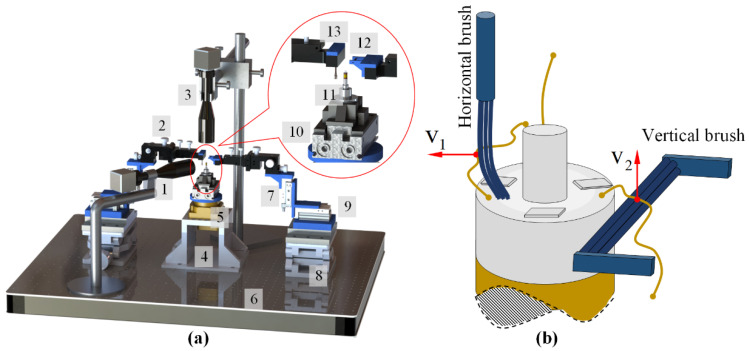
Setup of automated wires in the S&A brushing micro-manipulation system: (**a**) The automated system of wires used in S&A brushing micro-manipulation. (**b**) Flexible brush mechanism, with the following characteristics: 1—vision system A, 2—adjustment mechanism with four degrees of freedom, 3—vision system B, 4—support base, 5—rotating stage, 6—optical platform, 7—slide cylinder A, 8—vertical lifting platform, 9—slide cylinder B, 10—finger cylinder, 11—welding clamp, 12—vertical brushing micro-manipulator, 13—horizontal brushing micro-manipulator.

**Figure 4 micromachines-12-01140-f004:**
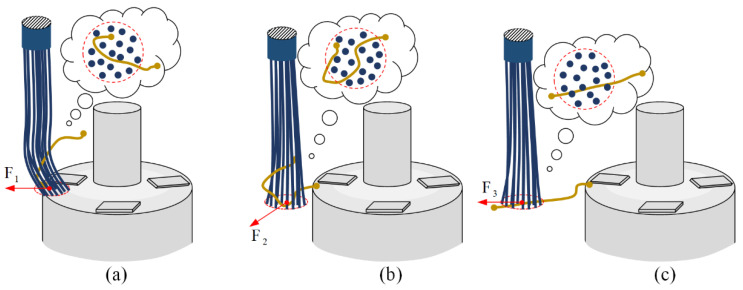
Three states in the working process of the horizontal brush: (**a**) The state of the horizontal brush in contact with the top of the winding. (**b**) The instantaneous state of the horizontal brush off the top of the winding. (**c**) The state of the horizontal brush with unfolded wire.

**Figure 5 micromachines-12-01140-f005:**
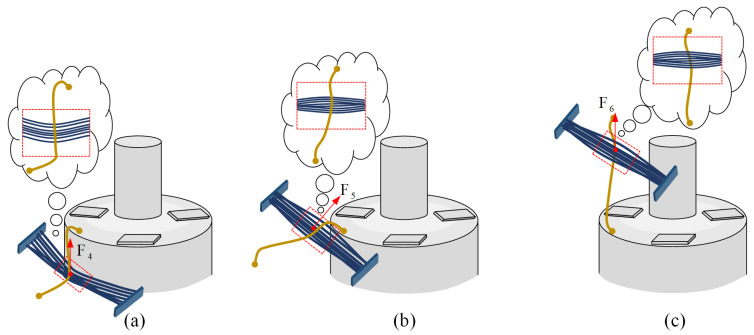
Three states in the working process of the vertical brush: (**a**) The state of the vertical brush in contact with the winding sidewall. (**b**) The instantaneous state of the vertical brush off the surface of the winding sidewall. (**c**) The state of the vertical brush with an adjusted wire attitude.

**Figure 6 micromachines-12-01140-f006:**
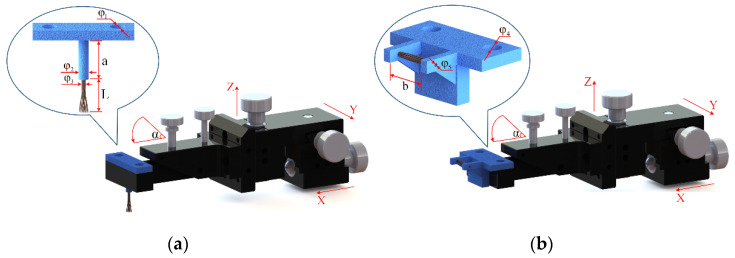
Setup of brushing micro-manipulator: (**a**) Horizontal brushing micro-manipulator. (**b**) Vertical brushing micro-manipulator.

**Figure 7 micromachines-12-01140-f007:**
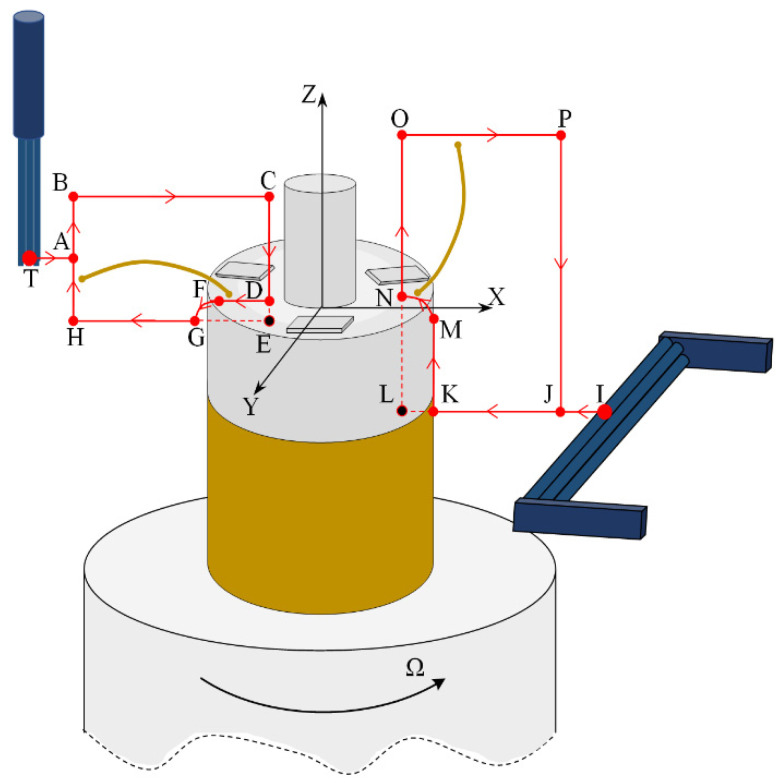
The motion trajectory of the brushing micro-manipulator.

**Figure 8 micromachines-12-01140-f008:**
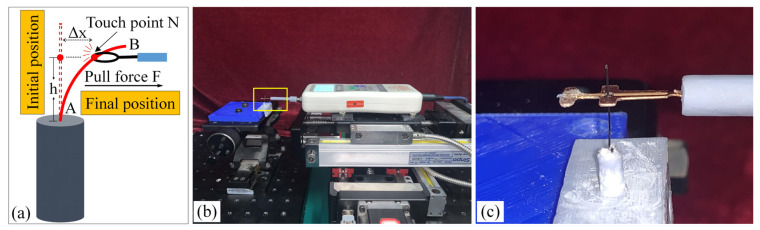
Setup of experimental system for the brush elastic force measurement: (**a**) The principle of the force measurement experiment. (**b**) The force measurement experimental system. (**c**) An enlarged view of the area marked by the yellow box, and it shows the contact part between the clamp and the brush.

**Figure 9 micromachines-12-01140-f009:**
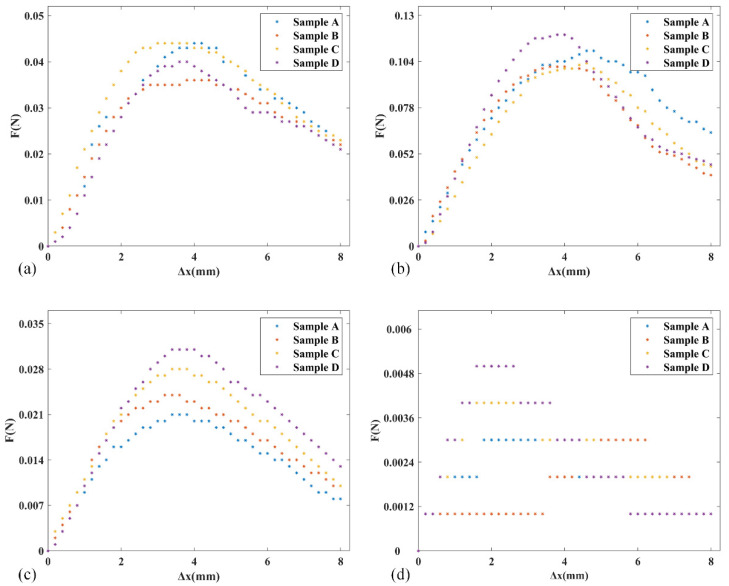
The relationship curves between *F*, Δ*x* and four different brush samples: (**a**) The relationship between *F* and Δ*x* for four brush samples of nylon material. (**b**) The relationship between *F* and Δ*x* for four brush samples of bristle material. (**c**) The relationship between *F* and Δ*x* for four brush samples of nanometer-silk material. (**d**) The relationship between *F* and Δ*x* for four brush samples of wool material.

**Figure 10 micromachines-12-01140-f010:**
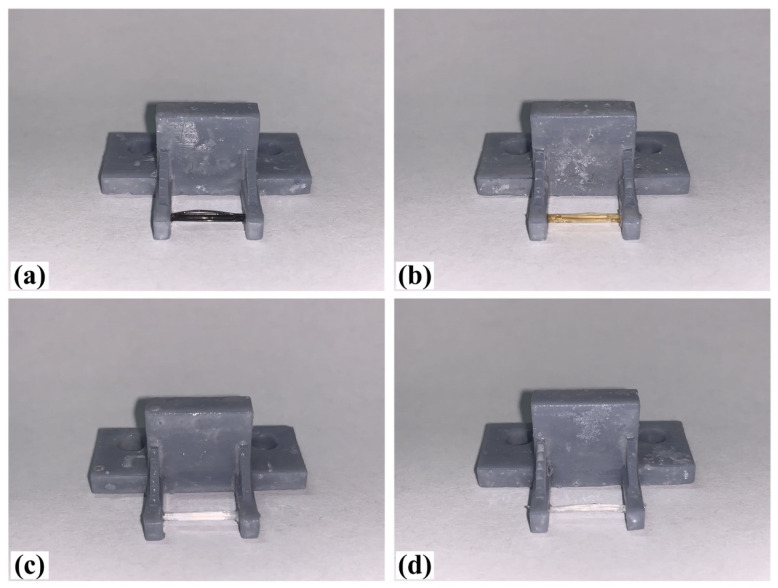
Vertical brushing micro-manipulators made of different materials: (**a**) nylon; (**b**) bristles; (**c**) nanometer-silk; (**d**) wool. All four brushes are 1mm in diameter and 10mm in length.

**Table 1 micromachines-12-01140-t001:** Geometric dimensions of the coreless motor’s winding in Figure 1a.

Parameter	*Φ*_1_ (mm)	*Φ*_2_ (mm)	*Φ*_3_ (mm)	*H* (mm)	*L*_1_ (mm)
Value	0.05~0.07	2~3	1	11~12	1.5~2

**Table 2 micromachines-12-01140-t002:** Geometric dimensions of brushing micro-manipulator.

Parameter	*φ*_1_ (mm)	*φ*_2_ (mm)	*φ*_3_ (mm)	*φ*_4_ (mm)	*φ*_5_ (mm)	*a* (mm)	*L* (mm)	*b* (mm)	*α* (°)
Value	4.5	3	1	4.5	1	13	3~10	10	5

**Table 3 micromachines-12-01140-t003:** The mean value of Δ*x* and *F* for brushes of different materials in Figure 8.

Material	Nylon	Bristles	Nanometer Silk	Wool
Mean(Δ*x*) (mm)	3.7500	4.2000	3.5500	3.1000
Mean(*F*) (N)	0.0410	0.1083	0.0260	0.0038

**Table 4 micromachines-12-01140-t004:** Brushing Micro-manipulators made of different materials and lengths.

Material for Brush	Brush with *L* = 3 mm	Brush with *L* = 5 mm	Brush with *L* = 7 mm	Brush with *L* = 10 mm
Nylon	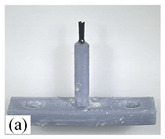	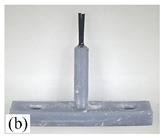	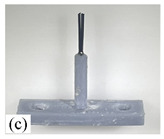	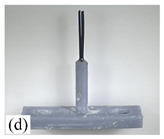
Bristles	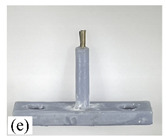	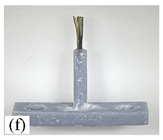	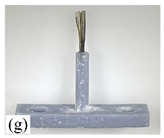	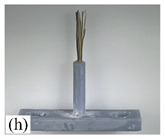
Nanometer-silk	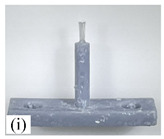	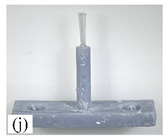	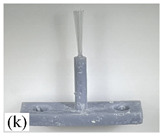	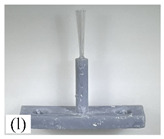
Wool	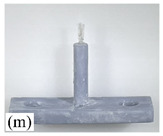	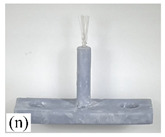	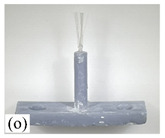	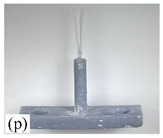

**Table 5 micromachines-12-01140-t005:** Subjective evaluation criteria of brushing result.

Level	Evaluation	Req1_H	Req2_H	Req3_H
1	Excellent	All conformity	All conformity	Conformity
2	Good	All conformity	All conformity	Not conformity
3	Intermediate level	All conformity	Partial conformity	Not conformity
4	Pool	Partial conformity	Partial conformity	Not conformity
5	Unacceptable	None conformity	None conformity	Not conformity

In the above table, “All”, “Partial” and “None” before “conformity” refers to the number of electrode wires; “All conformity” means that all three wires meet the requirements. ”Partial conformity” means that one or both wires meet the requirements; “None conformity” means that no wires meet the requirements.

**Table 6 micromachines-12-01140-t006:** Brushing results using the micro-manipulators listed in Table 4 made by nylon and bristle material.

Brush	Initial Side View	Initial Top View	Wire A	Wire B	Wire C
Brush (a)	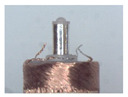	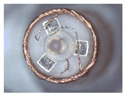	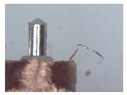	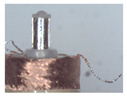	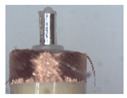
Brush (b)	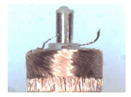	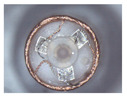	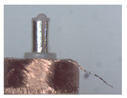	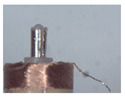	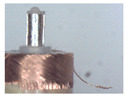
Brush (c)	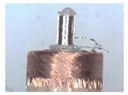	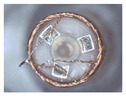	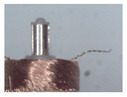	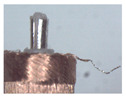	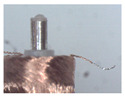
Brush (d)	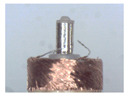	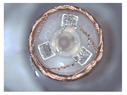	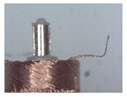	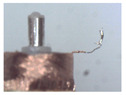	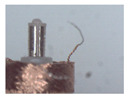
Brush (e)	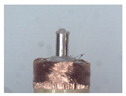	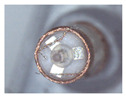	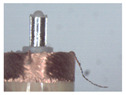	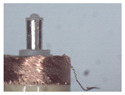	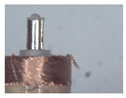
Brush (f)	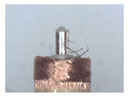	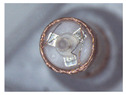	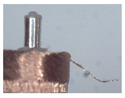	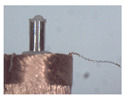	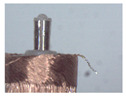
Brush (g)	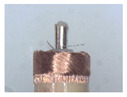	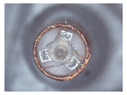	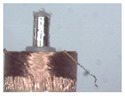	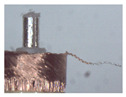	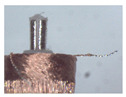
Brush (h)	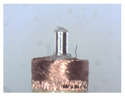	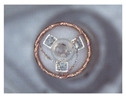	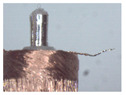	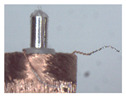	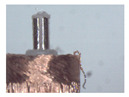

**Table 7 micromachines-12-01140-t007:** Brushing results using micro-manipulators in Table 4 made by nanometer-silk and wool material.

Brush	Initial Side View	Initial Top View	Wire A	Wire B	Wire C
Brush (i)	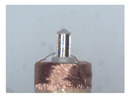	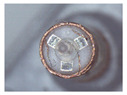	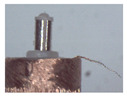	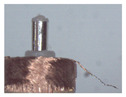	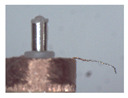
Brush (j)	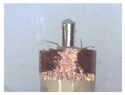	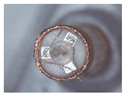	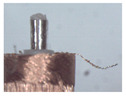	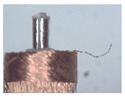	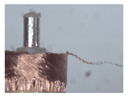
Brush (k)	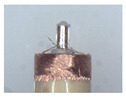	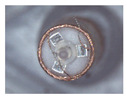	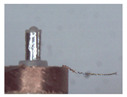	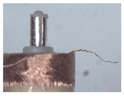	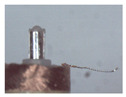
Brush (l)	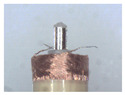	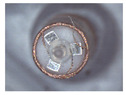	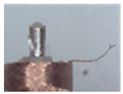	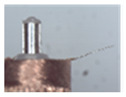	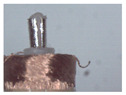
Brush (m)	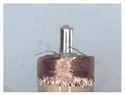	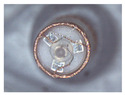	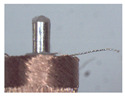	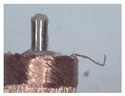	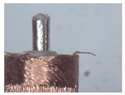
Brush (n)	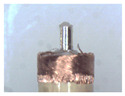	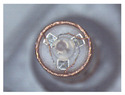	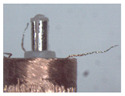	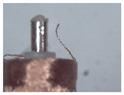	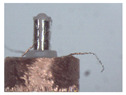
Brush (o)	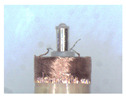	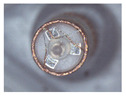	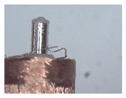	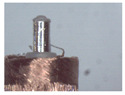	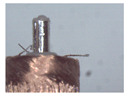
Brush (p)	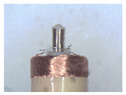	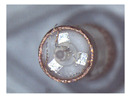	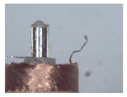	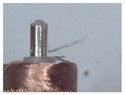	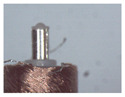

**Table 8 micromachines-12-01140-t008:** Subjective evaluation of brushing results for sixteen horizontal micro-manipulators.

Brush	Evaluation Level	Brush	Evaluation Level	Brush	Evaluation Level	Brush	Evaluation Level
Brush (a)	4	Brush (e)	5	Brush (i)	2	Brush (m)	4
Brush (b)	3	Brush (f)	4	Brush (j)	1	Brush (n)	4
Brush (c)	2	Brush (g)	3	Brush (k)	1	Brush (o)	5
Brush (d)	4	Brush (h)	3	Brush (l)	2	Brush (p)	5

**Table 9 micromachines-12-01140-t009:** Brushing results for two types of samples using the micro-manipulators shown in Figure 10.

Brush	Sample Type	Initial Side View	Initial Top View	Wire A	Wire B	Wire C
Brush (a)	SW	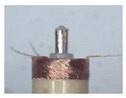	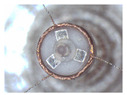	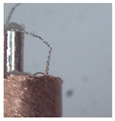	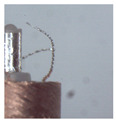	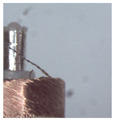
Brush (b)	SW	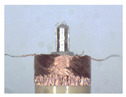	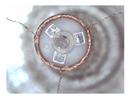	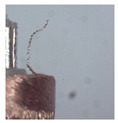	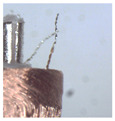	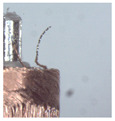
Brush (c)	SW	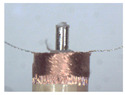	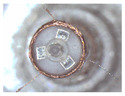	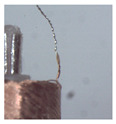	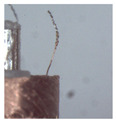	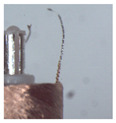
Brush (d)	SW	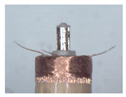	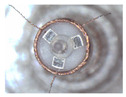	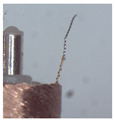	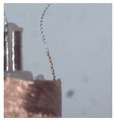	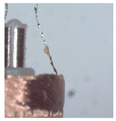
Brush (a)	AW	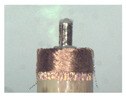	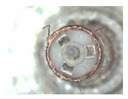	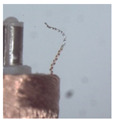	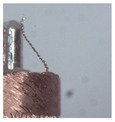	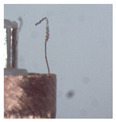
Brush (b)	AW	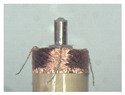	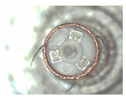	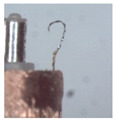	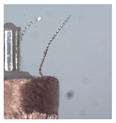	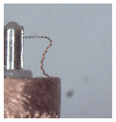
Brush (c)	AW	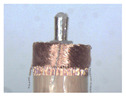	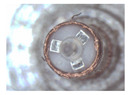	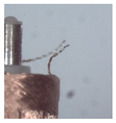	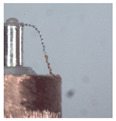	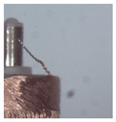
Brush (d)	AW	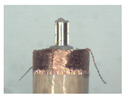	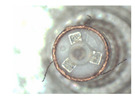	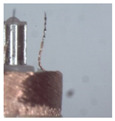	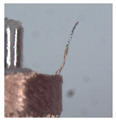	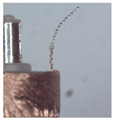

SW = Stretching wire; AW = Attaching wire.

**Table 10 micromachines-12-01140-t010:** Subjective evaluation of brushing results for four vertical micro-manipulators.

Brush	Sample Type-SW				Sample Type-AW			
	Brush(a)	Brush(b)	Brush(c)	Brush(d)	Brush(a)	Brush(b)	Brush(c)	Brush(d)
Evaluation level	4	4	1	1	4	4	3	2

**Table 11 micromachines-12-01140-t011:** Brushing results under the composition mode of horizontal brushing and vertical brushing.

Code	Winding_1	Winding_2	Winding_3	Winding_4	Winding_5	Winding_6
Side view1	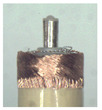	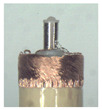	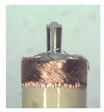	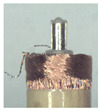	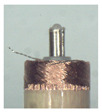	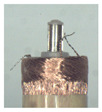
Top view1	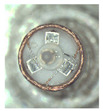	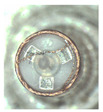	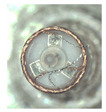	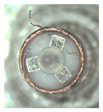	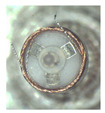	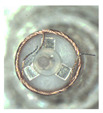
Side view2	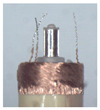	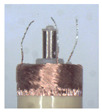	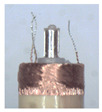	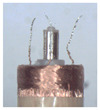	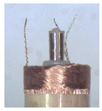	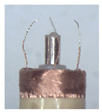
Top view2	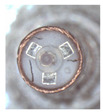	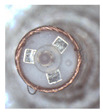	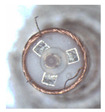	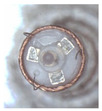	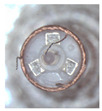	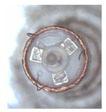

In the above table, ‘view1’ and ‘view2’ represent the status before and after brushing.

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
