# Peer review of "Design and Experimental Investigations of Shape and Attitude Carding System for the Wires of Micro Coreless Motor Winding"

_micromachines, 2021, doi:10.3390/mi12101140_

Round 1

Reviewer 1 Report

  1. The topic addressed in the paper is of current interest and in line with the profile of the journal.
  2. The paper describes in detail the results of research developed by the authors in connection with correcting the form of the wire using micromanipulators.
  3. In the Abstract, the wording in rows 17 and 18 is less common. One could write “The experimental results show that the material of brushing micro-manipulator has a great impact on the carding quality.”, instead of “The experimental results show that: The material of brushing micro-manipulator has a great impact on the carding quality. ”.
  4. In row 183, there is a reference to Req2, but this has not been previously defined.
  5. The authors could pay more attention to editing the article. It is recommended that a correction of the expression in English be made by a person who knows this language better. Some observations in this regard are presented below.

It is less common to include explanations between braces type { } (rows 134, 135, etc.). It could be written “(Fig. 1d- Fig. 1e)” instead of “{Fig. 1 (d) → Fig. 1 (e)} ”, etc.

The formulation “Sun et al. [18] Aiming at the spin problem of artificial satellite, the derotation mechanism, a flexible deceleration 86 brush, is proposed. ” is confusing.

In row 109, it can write "in" instead of "In".

In row 119, it can write "processes" instead of "process".

In the text concerning Table 1, it should be noted that the size symbols are those in Figure 1. It is common for the letters in the size symbols to be written in italics. A space can be left between the size symbols and the units of measurement or the parentheses in which the units of measurement were inscribed (in rows 544, 674, 675, in table 4, etc.). A free space must also be placed in row 517, in the case of the sequence “brushes{Brush(a)-(d)}” and respectively after the comma before the word “only”. Free spaces are still needed in rows 527, 531, 535, 620, etc.

In rows 429-430, the running speed should probably be expressed in mm/s and not in mm/s2. In line 431 it could write "pixels" instead of "pixel".

The explanation placed immediately below table 5 (“Where‘ All ’,‘ Partial ’and‘ None ’refers to the number of electrode wires”) could be written as an independent sentence in the last line of the table.

In row 566, a dot could be placed instead of the semicolon (“criterion; Of the four”). The observation is also valid in other lines of the paper (for example, row 609, etc.).

In row 678 it can be written “carding. The results”, instead of “carding . The results” (change the spaces before and after the dot).

In row 434, the value of the angle must be expressed in º and not in O

The characters used in Figure 8a are too small.

After the last word of the title of a table, a dot could be placed.

Sometimes capital letters were used in a sentence, without this being necessary (“Point” in line 370, etc.). In other cases, lowercase letters were used, although uppercase letters were needed (in line 395 write “. Point I is located below”, instead of “. point I is located below”.

The list of bibliographic references needs to be drawn up more carefully. Thus, it is not advisable to use the expression "et al." instead of the names of some authors of the cited works. The paper authors did not use italics to write the titles of the journals and the volume number, they did not use bold characters to write the year of publication, they wrote "PP." instead of "pp." in row 750, etc. In the case of reference no. 5, only the first letter of the title of the cited paper must be written as a capital letter.

Reviewer 2 Report

  1. Abstract: The abstract should explain the novel contribution regarding previous contributions reported in the literature. What is the novel contribution of this paper?
  2. Introduction: In the introduction part, several scientific articles on similar topics reported in the literature should be cited.
  3. The text of the article must be corrected from the point of view of English grammar. For example, in line 124, the first word of the sentence “the” must begin with a capital letter.
